# REUSE AND DIFFUSE: ITERATIVE DENOISING FOR TEXT-TO-VIDEO GENERATION

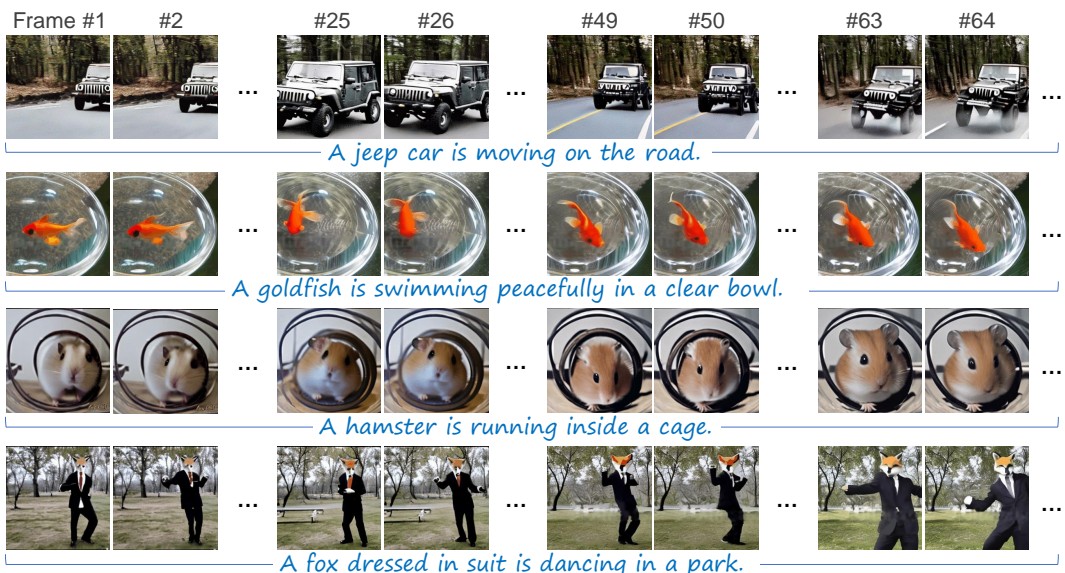

Figure 1: These are example videos with diverse content, generated using VidRD, guided by the text prompts below. With a diffusion model for video synthesis, video frames can be generated iteratively by reusing noise and imitating the diffusion process clip by clip. A large number of frames can be finally generated, and smoothness across frames can also be maintained.

## ABSTRACT

Inspired by the remarkable success of Latent Diffusion Models (LDMs) for image synthesis, we study LDM for text-to-video generation, which is a formidable challenge due to the computational and memory constraints during both model training and inference. A single LDM is usually only capable of generating a very limited number of video frames. Some existing works focus on separate prediction models for generating more video frames, which suffer from additional training cost and frame-level jittering, however. In this paper, we propose a framework called "Reuse and Diffuse" dubbed *VidRD* to produce more frames following the frames already generated by an LDM. Conditioned on an initial video clip with a small number of frames, additional frames are iteratively generated by reusing the original latent features and imitating the previous diffusion process. Besides, for the autoencoder used for translating between pixel space and latent space, we inject temporal layers into its decoder and fine-tune these layers for higher temporal consistency. We also propose a set of strategies for composing video-text data that involve diverse content from multiple existing datasets including video datasets for action recognition and image-text datasets. Extensive experiments show that our method achieves good results in both quantitative and qualitative evaluations. Our project page is available at https://anonymous0x233.github.io/ReuseAndDiffuse/.

# 1 INTRODUCTION

Text-to-video synthesis (Esser et al., 2023; Blattmann et al., 2023; Ge et al., 2023) recently has become an increasingly popular research topic in the field of Artificial Intelligence Generated Content (AIGC) following the success of Diffusion Models for image synthesis (Rombach et al., 2022). This technique allows businesses to create engaging videos from written text without the need for expensive equipment or professional illustrators. Video creation will become more efficient and innovative with the advancement of artificial intelligence technologies.

Existing video synthesis methods have achieved some progress, but the quality of the generated videos remains less than satisfactory. On the one hand, improving temporal consistency while generating diverse content remains a big challenge. On the other hand, a typical LDM is only capable of generating a few video frames due to the limitation of computation and memory resources. High-quality smooth videos, involving diverse content and containing a quantity of frames, are preferable in real applications. For this purpose, previous works such as FDM (Harvey et al., 2022), MCVD (Voleti et al., 2022) and Video LDM (Blattmann et al., 2023), exploit prediction mechanisms for producing future frames based on current video frames. Frame prediction, however, is no easier than direct video generation, and solving frame-level jittering is difficult. Furthermore, a cascaded pipeline, involving a video generation module and a prediction module, introduces more training cost and inference time. In this paper, we propose a novel framework called "Reuse and Diffuse" dubbed *VidRD* and Figure 1 shows some examples generated by it. VidRD can generate more coherent and consistent video frames by leveraging the previous frames generated by a single LDM. After generating an initial video clip by LDM, the following frames are produced iteratively by reusing the latent features of the previous clip and imitating the previous diffusion process. VidRD contains a temporal-aware LDM based on a pre-trained LDM for image synthesis. To train our model efficiently, we initialize the parameters of our spatial layers with a pre-trained image LDM. We also reform and fine-tune the decoder of autoencoder by injecting temporal layers into it. For iterative generation, VidRD contains three novel modules: Frame-level Noise Reversion (FNR), Past-dependent Noise Sampling (PNS), and Denoising with Staged Guidance (DSG). FNR reuses the initial noise in reverse order from the previous video clip, while PNS brings a new random noise for the last several video frames. Furthermore, temporal consistencies between video clips are refined by DSG.

Moreover, the training of LDMs usually relies on a massive amount of data to ensure the quality of the generative content (Khachatryan et al., 2023). The scarcity of high-quality video-text data has always been a problem. To this end, we devise a set of strategies to utilize existing datasets including video datasets for action recognition and image-text datasets. In addition to the typical video datasets in which each video is captioned with a short descriptive sentence, we use multi-modal Large Language Models (LLMs) to segment and caption videos in action recognition video datasets. Additionally, images with text captions are transformed into pseudo-videos by random zooming and panning so visual content of videos can be largely enriched.

Extensive experiments demonstrate that VidRD consistently achieves high performance in both quantitative and qualitative evaluations. On the benchmark on UCF-101 (Soomro et al., 2012), we achieve Fréchet Video Distance (FVD) of 363.19 and Inception Score (IS) of 39.37.

In summary, our contributions are three-fold:

- We propose an iterative text-to-video generation method that leverages a temporal-aware LDM to generate smooth videos. By reusing the latent features of the already generated video clip and imitating the previous diffusion process every time, more video frames can be produced iteratively.
- A set of effective strategies is proposed to compose a high-quality video-text dataset. We use LLMs to segment and caption videos from action recognition datasets. Image-text datasets are also used by transforming into pseudo-videos with random zooming and panning.
- Extensive experiments on UCF-101 benchmark demonstrate that VidRD achieves good FVD and IS in comparison with existing methods. Qualitative evaluations also show good results.

# 2 RELATED WORK

**Image synthesis models.** Automatic image synthesis is seen as a major milestone towards general artificial intelligence (Goertzel & Pennachin, 2007; Clune, 2019; Fjelland, 2020; Zhang et al.,

2023). Diffusion Models (DMs) show amazing results for text-guided image synthesis. Massive works (Kawar et al., 2023; Bhunia et al., 2023; Liu et al., 2023b; Fan et al., 2023; Dabral et al., 2023; Huang et al., 2023; Ramesh et al., 2022; Saharia et al., 2022) focus on this technology. GLIDE (Nichol et al., 2021) and Stable Diffusion (Rombach et al., 2022) are two representative DM-based works that employ Vision-Language Models (VLMs) such as CLIP (Radford et al., 2021) for text-guided image synthesis. With the rapid development of DMs, advanced image editing task has also been achieved.

**Video synthesis models.** Recently, motivated by DM-based image synthesis, several works (Esser et al., 2023; Ge et al., 2023; Blattmann et al., 2023; Khachatryan et al., 2023; He et al., 2022; Dabral et al., 2023; Luo et al., 2023; Brooks et al., 2022) propose to explore DMs for conditional video synthesis. Among these works, Video LDM (Blattmann et al., 2023) is a representative work and also exhibits excellent results. On the basis of an LDM for image synthesis pre-trained on large-scale image-text data, Video LDM fine-tunes its newly added temporal layers with video data. In addition, the authors propose an interpolation model and an upsampler model for generating high-quality videos. Almost at the same time, PYoCo (Ge et al., 2023) is proposed as an improved method for extending LDM from image synthesis to video synthesis. Based on the continuity of video content over time, the authors design a video noise prior to achieve better temporal consistency. With the satisfactory results of LDM-based video synthesis, there are also some works (Molad et al., 2023; Qi et al., 2023; Liu et al., 2023a) on controllable video editing.

## 3 PRELIMINARIES

### 3.1 LATENT DIFFUSION MODELS

DMs learn to model a data distribution $p_{\text{data}}$ via iterative denoising from a noise distribution so the desired data distribution can be generated. Given samples $\mathbf{x_0} \sim p_{\text{data}}$, the diffusion forward process iteratively adds noise:

$$q(\mathbf{x_t} \mid \mathbf{x}_{t-1}) = \mathcal{N}(\mathbf{x}_t; \alpha_t \mathbf{x}_{t-1}, \sigma_t^2 \mathbf{I}) \tag{1}$$

which represents the conditional density of $\mathbf{x}_t$ given $\mathbf{x}_{t-1}$. Here, a noise schedule is defined by $\alpha_t$ and $\sigma_t$ parameterized by diffusion time $t$. For generating a fully random noise with the increase of diffusion time $t$, signal-to-noise ratio $\lambda_t = \log(\alpha_t^2/\sigma_t^2)$ needs to monotonically decrease. To this end, a variance-preserving time schedule satisfying $\alpha_t^2 + \sigma_t^2 = 1$ is usually used. Following the closure of normal distribution, we can directly sample $\mathbf{x}_t$ at any diffusion time $t$ by:

$$q(\mathbf{x_t} \mid \mathbf{x}_0) = \mathcal{N}(\mathbf{x}_t; \bar{\alpha}_t \mathbf{x}_0, (1 - \bar{\alpha}_t^2)\mathbf{I}) \tag{2}$$

where $\bar{\alpha}_t = \prod_{i=1}^t \alpha_i$.

In the backward process of diffusion, a model denoted by $f_\theta$ parameterized by $\theta$ is trained to predict the noise to iteratively recover $\mathbf{x}_0$ from $\mathbf{x}_T$ which is noisy data after adding noise $T$ times. As long as $T$ is large enough, the original sample of real data is fully perturbed such that $\mathbf{x}_T \sim \mathcal{N}(\mathbf{0}, \mathbf{I})$. While training, the denoising matching score is optimized following:

$$\mathbb{E}_{\mathbf{y} \sim \mathcal{N}(\mathbf{0}, \mathbf{I})}[\|\mathbf{y} - f_\theta(\mathbf{x}_t; \mathbf{c}, t)\|_2^2] \tag{3}$$

where $\mathbf{y}$ representing the target features can be a random noise and $\mathbf{c}$ is an optional conditioning signal such as text prompt in text-to-something DMs. Also, $t$ is sampled from a uniform distribution which is set to $\mathcal{U}\{0, 1000\}$ in Stable Diffusion (Rombach et al., 2022). Once $f_\theta$ is trained, we can generate a novel $\mathbf{x_0}$ from a random noise $\mathbf{x}_T \sim \mathcal{N}(\mathbf{0}, \mathbf{I})$ with a deterministic sampling DDIM (Song et al., 2021).

Since training DMs in pixel space requires a large amount of computational resources, Stable Diffusion (Rombach et al., 2022) proposes to apply a regularized autoencoder to compress the original pixels into latent space to save computation and memory. In this way, DMs are transformed into Latent Diffusion Models (LDMs). The autoencoder in an LDM consists of an encoder $\mathcal{E}$ for encoding pixel features $\mathbf{x}$ into latent features $\mathbf{z}$ and a decoder $\mathcal{D}$ for decoding $\mathbf{z}$ back to $\mathbf{x}$. In general, the autoencoder is trained by reconstructing:

$$\hat{\mathbf{x}} = \mathcal{D}(\mathcal{E}(\mathbf{x})) \approx \mathbf{x} \tag{4}$$

where $\hat{\mathbf{x}}$ denotes the reconstructed sample after the real data $\mathbf{x}$ is processed by the encoder and the decoder in turn. For the typical implementation of the autoencoder for an image LDM such as Stable Diffusion, both the encoder $\mathcal{E}$ and the decoder $\mathcal{D}$ are for static images only. For an LDM for video synthesis, it works frame by frame so no temporal information is considered.

## 4 METHOD

### 4.1 MODEL ARCHITECTURE

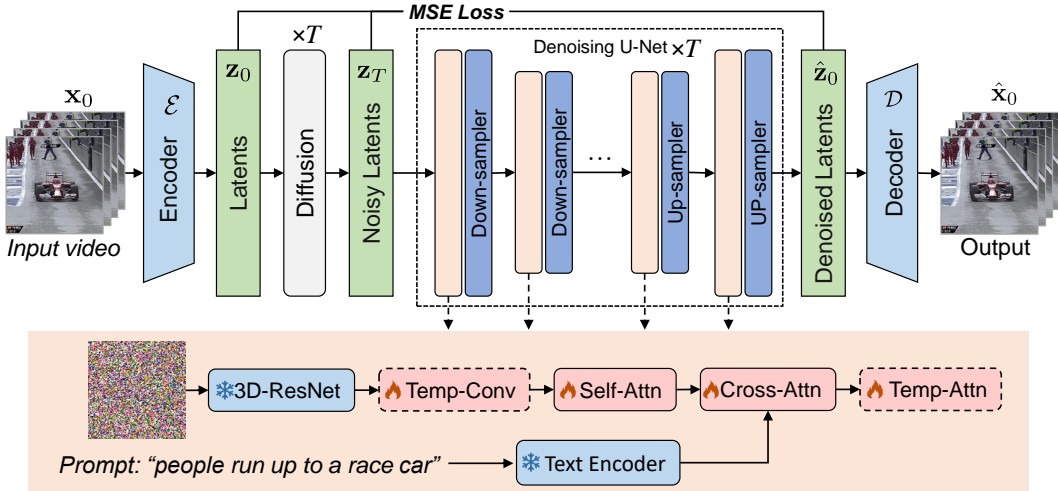

Figure 2: The architecture of VidRD is derived from an LDM for image synthesis. Modules with snowflake marks are frozen while those with flame marks are trainable. Modules with dashed boxes are added in addition to the original LDM for image synthesis.

VidRD is based on the pre-trained Stable Diffusion for image synthesis, including its Variational Auto-Encoder (VAE) for latent representation and U-Net for latent denoising. Figure 2 shows the architecture of VidRD. We adapt the original U-Net for image diffusion to video synthesis by injecting temporal layers. These two types of temporal layers are: *Temp-Conv* representing 3D convolution layers and *Temp-Attn* representing temporal attention layers. Also, most network layers, except for the newly added *Temp-Conv* and *Temp-Attn*, in our devised U-Net, are initialized with the pre-trained model weights of Stable Diffusion. The parameters of *Temp-Conv* and *Temp-Attn* are randomly initialized with the last layer zeroed and residual connections are also applied.

For efficient training, only part of our network layers are trainable. All the parameters of the text encoder are frozen. For U-Net, existing works use either a two-stage (Blattmann et al., 2023) or alternating (Ge et al., 2023) training scheme with image and video data. Essentially, they use image data for fine-tuning spatial layers and video data for training temporal layers, respectively. Instead of this manual training scheme, our U-Net is trained with pure video data in a unified way since the image data are transformed into pseudo-videos. Specifically, as illustrated in Figure 9, the network modules in U-Net with red background, including two newly added temporal layers and the spatial attention layers originally designed in LDM for image synthesis, are trainable. Furthermore, since the autoencoder is originally designed for image synthesis, temporal relations between video frames are not considered. To achieve a more accurate representation of videos in the output pixel space, we inject temporal layers implemented with 3D convolutions into the decoder $\mathcal{D}$ and fine-tune it with video data. The details are presented in Section 4.4.

### 4.2 VIDEO-TEXT DATA COMPOSITION

The training of an LDM for text-guided video synthesis requires a large amount of captioned videos. Large-scale well-captioned video datasets such as VATEX (Wang et al., 2019) or WebVid-2M (Bain et al., 2021) are very limited however. To compensate for the lack of high-quality video-text data, we propose a set of strategies for composing video-text data from different types of existing datasets other than well-captioned video-text datasets like VATEX (Wang et al., 2019) and WebVid-2M (Bain et al., 2021). Figure 3 illustrates the three types of datasets that we use: text-image datasets, short video classification datasets, and long video classification datasets.

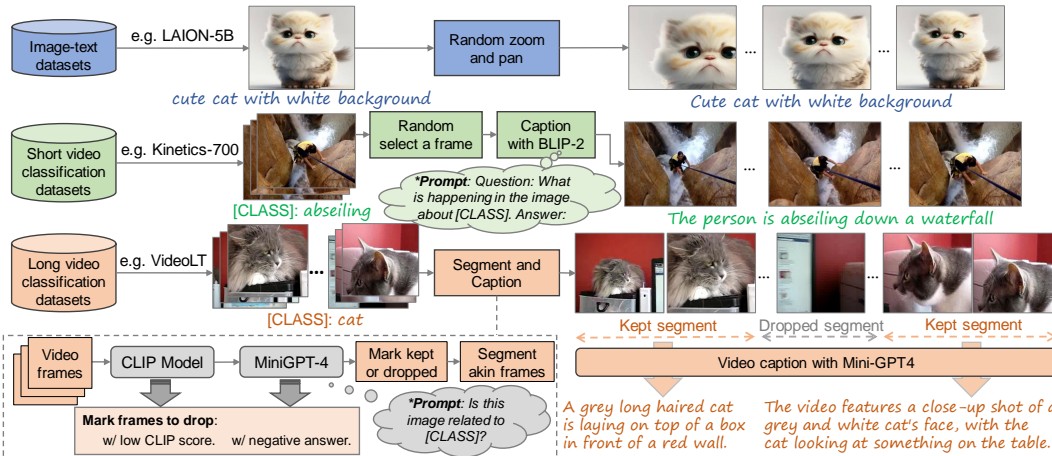

Figure 3: A set of strategies is devised for processing different types of datasets including image-text datasets, short video classification datasets, and long video classification datasets.

**Image-text datasets.** Since the quantity and quality of image datasets are much better than those of video datasets, it is necessary to exploit image datasets to improve the video generation model. As shown in the top part of Figure 3, for each image with a caption, we apply random zooming and panning to produce multiple images and they are further composed into a pseudo-video.

**Short video classification datasets.** For those short video datasets typically involving a single scene, the problem is how to give a proper text caption based on its classification label to each video. Our strategy is illustrated as green modules in Figure 3. For each short video with a given classification label such as "abseiling", we randomly select a video frame from it and then use BLIP-2 to generate a text caption by querying the LLM with the frame and its classification label. In order to make the LLM produce more diverse text captions, we use a few prompt templates for querying the LLM, and one of them is randomly selected every time.

**Long video classification datasets.** For the datasets containing long videos involving multiple scenes such as VideoLT (Zhang et al., 2021), it is improper to use the description of a single frame as the whole video's caption. To this end, we use a segment-then-caption strategy as shown in the bottom part of Figure 3. For each video, we employ CLIP (Radford et al., 2021) with vision-language alignment ability and MiniGPT-4 (Zhu et al., 2023) with vision-language understanding to mark those frames irrelevant to the classification label. Those frames with low CLIP matching scores with classification labels or considered irrelevant by MiniGPT-4 are marked to be dropped. A video can then be segmented with this attribute of each frame. To avoid producing too short videos, we drop those segments with too few frames. Finally, MiniGPT-4 is again used for captioning the segmented sub-videos with devised prompt templates.

## 4.3 LONGER VIDEO GENERATION

Since LDM for video generation requires a large amount of computation and memory, it is usually unable to generate a lot of frames at once. For this problem, we propose an iterative approach for longer video generation with a single LDM, and the whole pipeline is shown in Figure 4. We use $\mathbf{z}^{i,j}$ to represent the latent features of the $j$-th frame in the $i$-th video clip. Within each iteration, there are $N$ frames generated and the last $M$ frames are used as *prompt frames* for the next iteration. The whole process is briefly outlined in Algorithm 1. The details of the key modules are stated below.

**Frame-level Noise Reversion.** As previous works have revealed, for generating smooth videos, the initial noise of LDMs for video synthesis is essential (Ge et al., 2023), and sharing a base noise across video frames also helps (Luo et al., 2023). We borrow a similar scheme for generating longer videos. Specifically, the initial video clip is firstly generated with our trained LDM by denoising from an initial noise sampled from a normal distribution like:

$$\mathbf{z}_T^{0,j} \sim \mathcal{N}(\mathbf{0}, \mathbf{I}), \ \ j \in \{0, 1, \dots, N-1\} \tag{5}$$

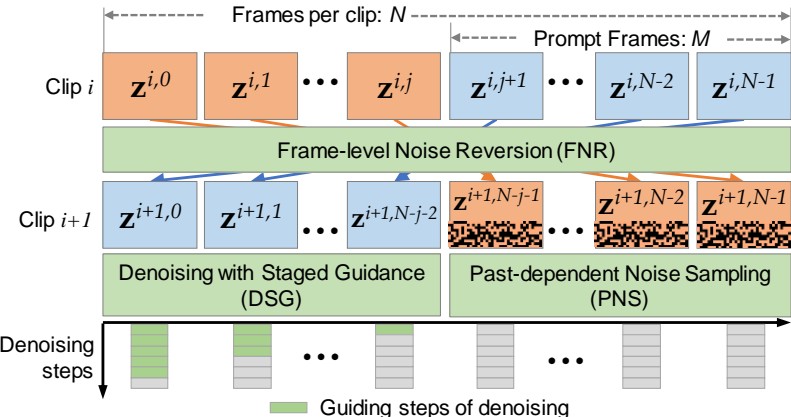

Figure 4: Videos can be generated iteratively with a single LDM. After each iteration, $N$ frames are generated and the last $M$ frames are used as prompt frames for the next iteration.

---

**Algorithm 1:** Iterative video generation.

---

**Input:** $\{\mathbf{z}_t^{0,j} \mid j \in \mathbb{Z} \cap [0, N), t \in \mathbb{Z} \cap [0, T]\}$: Latent features of the first clip.

**Output:** $\{\mathbf{z}_0^{i,j} \mid j \in \mathbb{Z} \cap [0, N), i \in \mathbb{Z} \cap [0, V_{\max})\}$: Denoised latent features of all $V_{\max}$ clips.

1  **for** $i = 1$ **to** $V_{max}$ **do**
2      **for** $j = 0$ **to** $N - 1$ **do**
3          $\mathbf{z}_T^{i,j} = \mathbf{z}_T^{i-1,N-j-1}$ ;       // Frame-level Noise Reversion (FNR)
4          **if** $j \geq M$ **then**        // Past-dependent Noise Sampling (PNS)
5              $\mathbf{z}_T^{i,j} = \mathbf{z}_T^{i,j}\alpha/\sqrt{1+\alpha^2} + \epsilon^{i,j};\ \ \epsilon^{i,j} \in \mathcal{N}(\mathbf{0}, \mathbf{I}/(1+\alpha^2))$;
6      **for** $t = T$ **to** $1$ **do**        // Denoising with Staged Guidance (DSG)
7          **if** $t > (1-\beta)T + \beta Tj/M$ **then** $\mathbf{z}_{t-1}^{i,j} = \mathbf{z}_{t-1}^{i-1,N-j-1}$;
8          **else** $\mathbf{z}_{t-1}^{i,j} = \text{DDIM}(\mathbf{z}_t^{i,j}, t)$    // Progressively denoising with DDIM;

---

where $N$ is the number of frames in a single video clip. To ensure continuity between video clips, FNR is proposed by reusing the initial noises in a reversed order every iteration. That is:

$$\mathbf{z}_T^{i,j} = \mathbf{z}^{i-1,N-j-1}, \ \ i \geq 1, j \in \{0, 1, \dots, N-1\} \tag{6}$$

In combination with Equation 5, the initial noise of each frame in the following video clips can be computed. However, FNR alone cannot guarantee that videos are smooth. The video content may simply become repetitive in some extreme cases. For this problem, PNS and DSG are proposed.

**Past-dependent Noise Sampling.** To mitigate the extent of video content repetition, which is critical to perception, randomness needs to be introduced on the basis of FNR. To this end, we propose Past-dependent Noise Sampling (PNS) which is used for introducing randomness gradually. Specifically, excluding the $M$ prompt frames, random noises are added to the remaining $N - M$ frames, which are initialized with that of $N - M$ frames of the previous video clip. Therefore, Equation 6 is modified based on the position of the frames in each video clip:

$$\mathbf{z}_T^{i,j} = \begin{cases} \mathbf{z}_T^{i-1,N-j-1} & \text{if } j < M \\ \frac{\alpha}{\sqrt{1+\alpha^2}}\mathbf{z}_T^{i-1,N-j-1} + \epsilon^{i,j} & \text{otherwise} \end{cases}, \ \ \epsilon^{i,j} \sim \mathcal{N}(\mathbf{0}, \frac{1}{1+\alpha^2}\mathbf{I}), \ \alpha \geq 0 \tag{7}$$

where $\epsilon^{i,j}$ is a newly added random noise, and $\alpha$ is a hyper-parameter for controlling the ratio of this noise to the original reversed noise. The results of PNS are identical to those of FNR when $j < M$ and the difference is only on the remaining $N - M$ frames. New random noise sampled from a standard normal distribution is used when $\alpha = 0$. A larger $\alpha$ brings more proportion of the reversed noise of the last video clip so higher temporal consistency can be achieved.

**Denoising with Staged Guidance.**

Simply reusing the initial noises are not enough for high temporal consistency so we propose Denoising with Staged Guidance (DSG) for replicating denoising process. The approach involves replicating some denoising steps by reusing the latent features from the DDIM process in generating the previous clip. At the same time, to avoid content repetition, a staged strategy for denoising with guidance is also used. Specifically, we have:

$$\mathbf{z}_{t-1}^{i,j} = \begin{cases} \mathbf{z}_{t-1}^{i-1,N-j-1} & \text{if } t > (1-\beta)T + \frac{\beta T j}{M} \\ \text{DDIM}(\mathbf{z}_t^{i,j}, t) & \text{otherwise} \end{cases}, \quad \beta \in [0,1] \tag{8}$$

where $\beta$ represents the extent of guided denoising. Latent features in the current clip are denoised totally with DDIM sampling when $\beta = 0$, and a larger $\beta$ brings more guidance for each latent feature with $j < M$. In this way, video content of the first $M$ prompt frames can be consistent with the last $M$ frames of the last clip, and new content can also emerge along with the denoising process because staged guidance becomes weaker as $j$ gets closer to $M$.

### 4.4 TEMPORAL-AWARE DECODER FINE-TUNING

Since the original autoencoder of Stable Diffusion is specifically designed for image synthesis, it is necessary to fine-tune it with video data for better performance of video synthesis. However, the latent features used as inputs to U-Net after encoding are critical for efficient training when the pre-trained weights of Stable Diffusion are loaded. The encoder remains unchanged and the weights are frozen during fine-tuning. Also, temporal relations across video frames need to be considered for better temporal consistency after decoding so we add *Temp-Conv* layers after *ResNet* of each block in the decoder. For efficient fine-tuning, only the newly added *Temp-Conv* layers are trainable. In addition, for better adapting from the autoencoder for image, we initialize the last layer of *Temp-Conv* with zero and apply a residual connection.

For fine-tuning the autoencoder, we use the same datasets used for training U-Net describe in Section 4.2. The fine-tuning also follows the adversarial manner following Stable Diffusion (Rombach et al., 2022). The total loss is as follows:

$$\mathcal{L} = \alpha_{\text{rec}}\mathcal{L}_{\text{rec}}(\mathbf{x}, \mathcal{D}(\mathcal{E}(\mathbf{x}))) + \alpha_{\text{reg}}\mathcal{L}_{\text{reg}}(\mathbf{x}; \mathcal{E}, \mathcal{D}) + \alpha_{\text{disc}}\mathcal{L}_{\text{disc}}(\mathcal{D}(\mathcal{E}(\mathbf{x}))) \tag{9}$$

In addition to the main reconstruction loss $\mathcal{L}_{\text{rec}}$ and a regularizing loss $\mathcal{L}_{\text{reg}}$ for regularizing the latent representation, a discrimination loss $\mathcal{L}_{\text{disc}}$ is also used which is computed by a patch-based discriminator for differentiating the original videos from the reconstructed. These three losses are respectively weighted with $\alpha_{\text{rec}}$, $\alpha_{\text{reg}}$ and $\alpha_{\text{disc}}$.

## 5 EXPERIMENTS

### 5.1 EXPERIMENTAL SETUPS

**Datasets for training.** In total, four types of datasets are used for training VidRD, including: **1)** Well-captioned video-text datasets: WebVid-2M (Bain et al., 2021), TGIF (Li et al., 2016), VATEX (Wang et al., 2019) and Pexels [1]; **2)** Short video classification datasets: Moments-In-Time (Monfort et al., 2021) and Kinetics-700 (Smaira et al., 2020); **3)** Long video classification datasets: VideoLT (Zhang et al., 2021); **4)** Image datasets: LAION-5B (Schuhmann et al., 2022). The statistics about these datasets and the model training details are provided in the Appendix.

**Evaluation metrics.** Following previous works like Make-A-Video (Singer et al., 2023), PY-oCo (Ge et al., 2023) and Video LDM (Blattmann et al., 2023), the following metrics for quantitative evaluation are used: *(i)* Fréchet Video Distance (FVD) (Unterthiner et al., 2019): Following Make-A-Video (Singer et al., 2023), we use a trained I3D model (Carreira & Zisserman, 2017) for calculating FVD. *(ii)* Inception Score (IS) (Saito et al., 2020): Following previous works (Singer et al., 2023; Hong et al., 2023; Blattmann et al., 2023), a trained C3D model (Tran et al., 2015) is used for calculating the video version of IS.

---

[1] https://huggingface.co/datasets/Corran/pexelvideos

## 5.2 MAIN RESULTS

To fully evaluate VidRD, we conduct both quantitative and qualitative evaluations. All the generated videos for evaluation are 16 frames in $256 \times 256$ resolution unless otherwise specified.

| Model | #Videos for Training | IS ↑ | FVD ↓ |
|---|---|---|---|
| CogVideo (Hong et al., 2023) | 5.4M | 25.27 | 701.59 |
| MagicVideo (Zhou et al., 2022) | 27.0M | - | 699.00 |
| LVDM (He et al., 2022) | 2.0M | - | 641.80 |
| ModelScope (Wang et al., 2023a) | - | - | 639.90 |
| Video LDM (Blattmann et al., 2023) | 10.7M | 33.45 | 550.61 |
| Make-A-Video (Singer et al., 2023) | 20.0M | 33.00 | 367.23 |
| VideoFactory (Wang et al., 2023b) | 140.7M | - | 410.00 |
| VidRD w/o fine-tuned VAE | 5.3M | 39.24 | 369.48 |
| VidRD w/ fine-tuned VAE | 5.3M | **39.37** | **363.19** |

Table 1: Quantitative evaluation results on UCF-101. All the videos for evaluation are generated in a zero-shot manner. In comparison with other methods, VidRD achieves better IS and FVD while using fewer videos for model training.

**Quantitative Evaluation.** Following previous works (Singer et al., 2023; Hong et al., 2023; Blattmann et al., 2023), we use UCF-101 (Soomro et al., 2012), a dataset for video recognition, for evaluating FVD and IS. Since there are only 101 brief class names such as *knitting* and *diving* in UCF-101, we devise a descriptive prompt for each class for video synthesis in our experiments. The whole list of prompts we use is provided in the Appendix. Following Make-A-Video (Singer et al., 2023), 10K videos are generated by VidRD following the same class distribution as the training set. The quantitative evaluation results are shown in Table 1. VidRD achieves the best FVD and IS while using much fewer videos for training. Meanwhile, fine-tuning VAE helps improve VidRD further. The reason is that a temporal-aware decoder can restore pixels from latent features more accurately.

**Qualitative Evaluation.** Since all the currently used metrics for evaluating video generation models are considered not fully reliable and may be inconsistent with perception, qualitative evaluation is necessary. To this end, example videos are generated with the same text prompts by VidRD and the other models including Make-A-Video (Singer et al., 2023), Imagen Video (Ho et al., 2022). Figure 5 shows the comparisons between the video generation results of these methods. VidRD performs well in both structure and appearance. More video examples can be found on our project website.

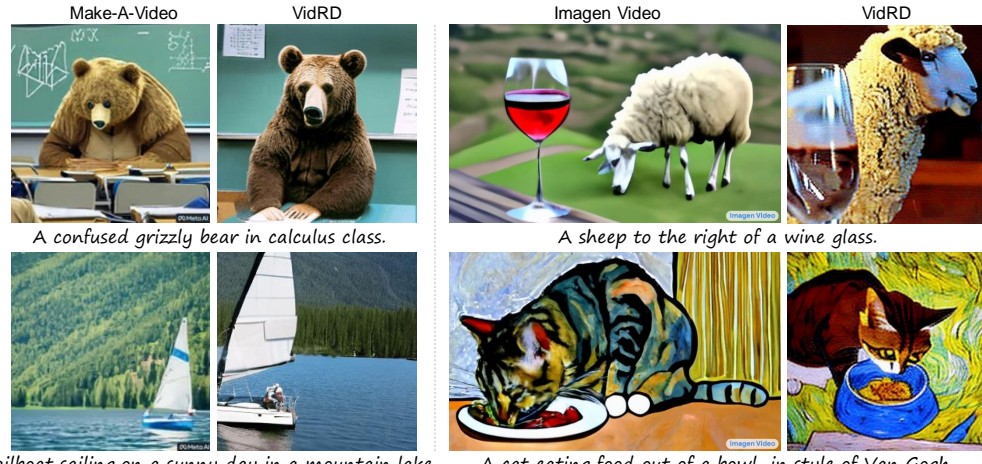

Figure 5: For comparison, some video examples generated by different methods are shown here. The examples generated by VidRD show good text alignment and structure.

## 5.3 ABLATION STUDY

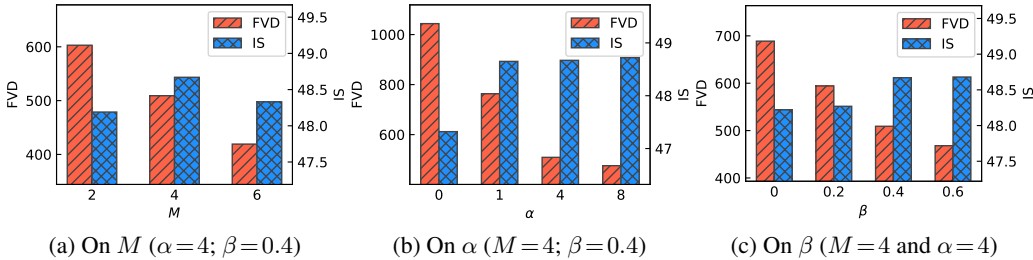

(a) On $M$ ($\alpha=4$; $\beta=0.4$)      (b) On $\alpha$ ($M=4$; $\beta=0.4$)      (c) On $\beta$ ($M=4$ and $\alpha=4$)

Figure 6: Ablation studies on hyper-parameters for a text-to-video generation under guidance scale set to 10 and the number of inference steps set to 50.

**Hyper-parameters for inference.** , we conduct ablation studies following the controlled variable method. The results are shown in Figure 6. Figure 6a reveals the parameter $M$ has a significant impact on temporal consistency but has little effect on structural quality. Figure 6b indicates that temporal consistency can be improved by reusing more noises from the previous clip. Also, the appearance or structure quality reflected by IS can be improved once the noise is reused, that is, $\alpha > 0$. Figure 6c shows that more guiding steps (larger $\beta$) help improve the output video quality.

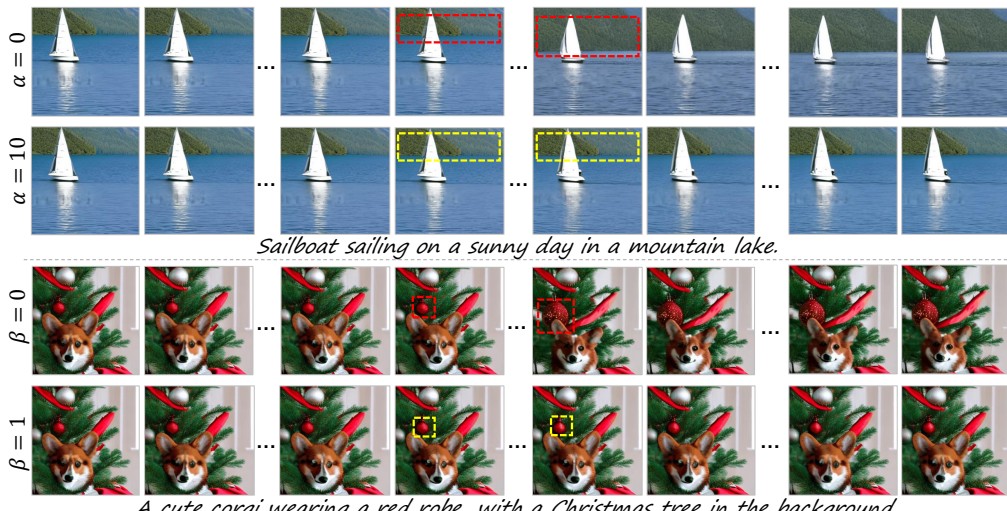

Figure 7: These are two visual examples showing the generated videos using different values of $\alpha$ and $\beta$ respectively. The red dashed box highlights the positions where abrupt, unreasonable visual changes occur. The yellow dashed box indicates that the issue is resolved by enabling PNS and DSG.

**Effects of PNS and DSG** To gain a more intuitive understanding of the parameters $\alpha$ and $\beta$, Figure 7 shows some generated videos with different parameter values. Some unreasonable abrupt video content changes, highlighted with red dashed boxes, can be easily observed when $\alpha = 0$ in which case no previous noises are reused. Similar artifacts can also be observed when $\beta = 0$ in which case there are no guiding steps for denoising the prompt frames.

## 6 CONCLUSION

In this work, we introduce a novel text-to-video framework called VidRD to generate smooth videos with text guidance. A set of strategies is proposed to exploit multiple existing datasets, which include video datasets for action recognition and image-text datasets, in order to train our text-to-video generation model. For generating longer videos, we propose an iterative approach through reusing the noise. and imitating the diffusion process clip-by-clip. Extensive experiments demonstrate that our method excels in both quantitative and qualitative evaluations.

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

# A APPENDIX

## A.1 MODEL DETAILS

### A.1.1 SAMPLING STRATEGY

To exploit the ability of image synthesis models, we use the pre-trained weights of Stable Diffusion v2.1 to initialize the spatial layers of our model. Both the VAE and the text encoder are frozen after they are initialized with pre-trained weights from Stable Diffusion. During model training, only the newly added temporal layers and transformer blocks of the spatial layers are trainable. Since our model is essentially an LDM, VAE of Stable Diffusion but with a fine-tuned decoder is used for latent representation. For LDM sampling, we use DDIM (Song et al., 2021) in all our experiments.

### A.1.2 TEMPORAL-AWARE VAE ARCHITECTURE

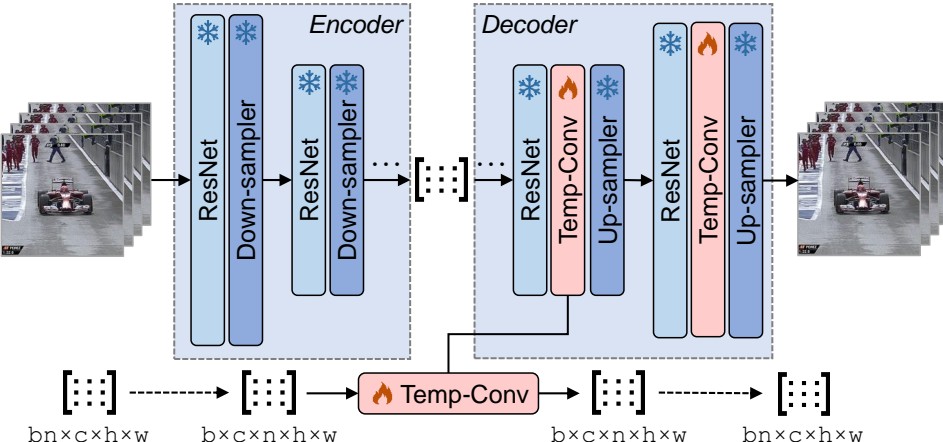

Figure 8: The backbone of the autoencoder is inherited from Stable Diffusion. *Temp-Conv* representing temporal layers and implemented with 3D convolutions are injected into the decoder. Most parts of the autoencoder are frozen and only the parameters of *Temp-Conv* are trainable.

To make the model restore pixels from latent features more accurately, temporal layers are injected into the decoder of VAE as Figure 8 shows.

### A.1.3 DETAILS OF TEMPORAL MODULES

We have input videos denoted by $\mathbf{x} \in \mathbb{R}^{B \times F \times 3 \times H' \times W'}$ where $B$, $F$, $H'$ and $W'$ respectively denote the inputs' batch size, number of frames, height, and width in the pixel space. After encoding, we can get its corresponding representation in the latent space $\mathbf{z} = \mathcal{E}(\mathbf{x}) \in \mathbb{R}^{B \times F \times C \times H \times W}$ where $C$, $H$ and $W$ respectively denote their channel, height, and width in the latent space.

For dealing with video inputs, the original 2D ResNet of Stable Diffusion is inflated to *3D-ResNet* by fusing the inputs' temporal dimension into batch dimension. In this way, this part of network parameters can be directly inherited from Stable Diffusion. For an input $\mathbf{z} = \mathcal{E}(\mathbf{x}) \in \mathbb{R}^{B \times F \times C \times H \times W}$, it is transformed into $\mathbb{R}^{(BF) \times C \times H \times W}$.

For strengthening temporal relations between video frames, two temporal modules are added. One temporal module is *Temp-Conv* implemented with 3D convolution layers which are added right after the ResNet block. Another temporal module is *Temp-Attn* which is an attention layer similar to *Self-Attn* in the original Stable Diffusion, but applied on the temporal dimension. Specifically, for *3D-ResNet*, *Temp-Conv* and *Temp-Attn*, the axes of input data are swapped accordingly to make the whole model work. The implementation details are shown in Figure 9.

## A.2 TRAINING DATASETS

We use three types of datasets for training and below are the details about these datasets:

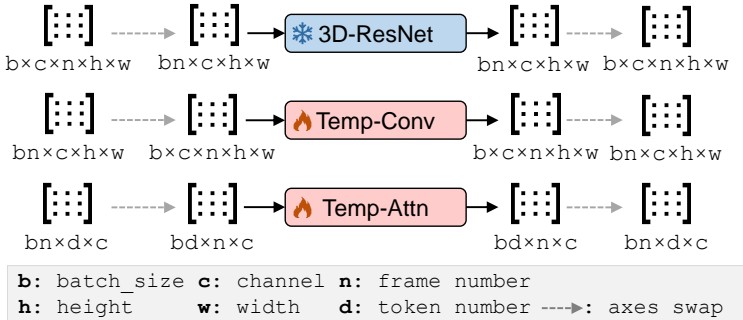

Figure 9: These are three essential network layers in VidRD. *3D-ResNet*, inherited from Stable Diffusion, treats the number of frames $n$ as a part of batch size. This is equivalent to applying the original *2D-ResNet* frame by frame so this layer is frozen in model training. *Temp-Conv*, implemented with 3D convolutions, processes video inputs in a tube manner while *Temp-Attn* applies attention layer along temporal axis.

1. **Well-captioned video-text datasets**: WebVid-2M (Bain et al., 2021), TGIF (Li et al., 2016), VATEX (Wang et al., 2019) and Pexels [2]. WebVid-2M contains a total of about 2.5 million subtitled videos but we only use those whose duration is less than 20 seconds. Additionally, we use a basic watermark removal solution to remove watermarks from the videos. TGIF consists of 100K GIFs collected from Tumblr and the duration is relatively short. VATEX is a large-scale well-captioned video datasets covering 600 fine-grained human activities. Pexels contains about 360,000 well-captioned videos from a popular website providing free stock videos.

2. **Short video classification datasets**: Moments-In-Time (Monfort et al., 2021) and Kinetics-700 (Smaira et al., 2020). Moments-In-Time contains more than one million videos covering 339 action categories and Kinetics-700 contains over 650,000 videos covering 700 human action categories. Most videos in these two datasets last a few seconds and each video is captioned using the strategy proposed in Section 4.2.

3. **Long video classification datasets**: VideoLT (Zhang et al., 2021). This dataset contains a total of 250,000 untrimmed long videos covering 1004 categories. After applying the strategy for long video classification datasets, we totally produce 800K captioned videos with an average length of 5 seconds.

4. **Image datasets**: LAION-5B. This dataset originally contains 5.58 billion image-text pairs but only a small part of it is used as compensation to our video-text data. After applying the strategy of transforming images to videos introduced in Section 4.2, 640K pseudo-videos with an average length of 2 seconds are produced.

| Dataset | WebVid-2M | TGIF | VATEX | Pexels | Moments-In-Time | Kinetics-700 | VideoLT | LAION-5B |
|---|---|---|---|---|---|---|---|---|
| Num. Videos (K) | 1,700 | 100 | 35 | 360 | 1,000 | 650 | 800 | 640 |
| Avg. Duration (s) | 11.9 | 3.1 | 10.0 | 19.5 | 3.0 | 10.0 | 5.0 | 2.0 |

Table 2: Four types of datasets are used including well-captioned video datasets, short video classification datasets, long video classification datasets, and image-text datasets.

Table 2 shows the statistics of the datasets for model training. VAE for encoding and decoding videos is the same as Stable Diffusion and only the newly added temporal layers in the decoder of VAE are trainable. For training the decoder of VAE, we set $\alpha_{rec}$, $\alpha_{reg}$ and $\alpha_{disc}$ to 1, $1^{-5}$ and 0.5 respectively. In U-Net, there are a total of 2.0B parameters, of which 565M are trainable and 316M are allocated for temporal layers. The base resolution of input videos for model training is $256 \times 256$ and 8 keyframes are sampled uniformly every 4 frames. The starting frame of sampling is randomly selected along the timeline. Each frame is resized along the shorter side and then randomly cropped to the target resolution. For video datasets with multiple captions such as VATEX (Wang et al., 2019),

---

[2]https://huggingface.co/datasets/Corran/pexelvideos

one caption is randomly chosen every time it is sampled. In general, these practices can be seen as data augmentation.

### A.3 MORE ABLATION STUDIES

#### A.3.1 CLASSIFIER-FREE GUIDANCE

For generating diverse videos, two hyper-parameters are critical in model inference: the number of inference steps and the scale of classifier-free guidance. The number of inference steps means the total steps of denoising from the initial random noises to the resulting latent features of video frames. The scale of classifier-free guidance is proposed with classifier-free diffusion guidance (Ho & Salimans, 2021). In each reversed diffusion step during model inference, the predicted noise $\tilde{\epsilon}_\theta$ is computed with two types of predictions: the prediction conditioned on prompt text features $\mathbf{c}$ that is $\epsilon_\theta(\mathbf{z}, \mathbf{c})$ and the prediction without such condition that is $\epsilon_\theta(\mathbf{z})$. The final predicted noise is calculated by combining these two, controlled with guidance scale $w$: $\tilde{\epsilon}_\theta = \epsilon_\theta(\mathbf{z}) + w(\epsilon_\theta(\mathbf{z}, \mathbf{c}) - \epsilon_\theta(\mathbf{z}))$. Classifier-free guidance is disabled when $w = 1$ and a larger $w$ means more video-text alignment but weaker diversity. To study the effects of these two hyper-parameters only, we do experiments using our base model without any strategies for iterative video generation. Figure 10 shows FVD and IS results of using different combinations of these hyper-parameters. The impact of the number of inference steps is relatively small and we set the number to 50 considering the efficiency of video synthesis. For the guidance scale, there is a trade-off between FVD and IS. A smaller guidance scale can achieve lower FVD which means higher temporal consistency. Yet a small guidance scale leads to low IS which means poor spatial appearance. Therefore, we set the guidance scale to $w = 10.0$.

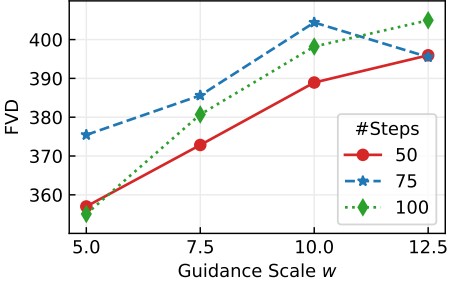
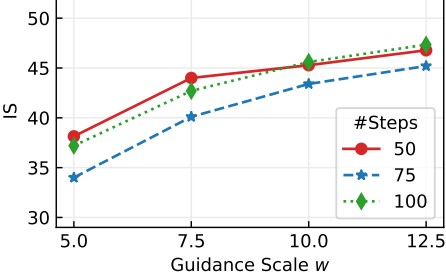

(a) FVD of using different guidance scale $w$ and the number of inference steps.

(b) IS of using different guidance scales $w$ and the number of inference steps.

Figure 10: Ablation results of the number of inference steps and the guidance scale are based on FVD and IS, which are evaluated on UCF-101 in a zero-shot manner.

#### A.3.2 JOINT TRAINING WITH IMAGE DATASETS

| Strategy | IS ↑ | FVD ↓ |
|---|---|---|
| VidRD w/o pseudo-videos | **42.00** | 451.16 |
| VidRD w/ pseudo-videos | 40.87 | **433.22** |

Table 3: This is a comparison between fine-tuning VidRD with and without pseudo-videos by randomly zooming and panning static images. FVD and IS are evaluated on UCF-101 and all experiments here are in a zero-shot manner.

To evaluate the effect of VidRD of joint training with pseudo-videos produced from image-text datasets, experiments are designed by fine-tuning VidRD with a small-scale dataset in which images are either in raw format or in pseudo-video style. This dataset totally consists of 5,000 videos from the VATEX (Wang et al., 2019) testing set and 8,000 images from LAION-5B (Schuhmann et al., 2022). We tabulate our findings in Table 3. Compared with using static images for training spatial layers only, we find that pseudo-videos, produced by random zooming and panning of static images, help enhance temporal consistency but compromise visual appearance. The reason is that pseudo-videos

bring more training for temporal layers while static images focus on spatial layers more. In practice, since the amount of video data is much less than that of image data, this technique is a cheap way to train a video diffusion model with high temporal consistency.

## A.4 PROMPTS FOR EVALUATION

For evaluating our method on UTF-101, we expand the original 101 class labels of UTF-101 as prompts. The arrows are flanked by the original class labels and the expanded prompts.

*apply eye makeup → applying eye makeup; apply lipstick → applying lipstick; archery → archery; baby crawling → baby crawling; balance beam → gymnast performing on a balance beam; band marching → band marching; baseball pitch → baseball pitcher throwing baseball; basketball → a basketball player shooting basketball; basketball dunk → dunking basketball in a basketball match; bench press → bench press; biking → biking; billiards → billiards; blow dry hair → blow dry hair; blowing candles → blowing candles; body weight squats → body weight squats; bowling → a person bowling on bowling alley; boxing punching bag → boxing punching bag; boxing speed bag → boxing speed bag; breast stroke → swimmer doing breast stroke; brushing teeth → brushing teeth; clean and jerk → clean and jerk; cliff diving → cliff diving; cricket bowling → bowling in cricket gameplay; cricket shot → batting in cricket gameplay; cutting in kitchen → cutting in kitchen; diving → diver diving into a swimming pool from a springboard; drumming → drumming; fencing → two fencers have fencing match indoors; field hockey penalty → field hockey match; floor gymnastics → gymnast performing on the floor; frisbee catch → group of people playing frisbee on the playground; front crawl → swimmer doing front crawl; golf swing → golfer swings and strikes the ball; haircut → haircutting; hammering → a person hammering a nail; hammer throw → an athlete performing the hammer throw; handstand pushups → an athlete doing handstand push up; handstand walking → an athlete doing handstand walking; head massage → massagist doing head massage to man; high jump → an athlete doing high jump; horse race → horse race; horse riding → group of people racing horse, person riding a horse; hula hoop → a woman doing hula hoop; ice dancing → man and woman dancing on the ice, ice dancing; javelin throw → athlete practicing javelin throw; juggling balls → a person juggling with balls; jumping jack → a young person doing jumping jacks; jump rope → a person skipping with jump rope; kayaking → a person kayaking in rapid water; knitting → knitting; long jump → an athlete doing long jump; lunges → a person doing lunges with barbell; military parade → military parade; mixing → mixing in the kitchen; mopping floor → mopping floor; nunchucks → a person practicing nunchuck; parallel bars → gymnast performing on parallel bars; pizza tossing → a person tossing pizza dough; playing cello → a musician playing the cello in a room; playing daf → a musician playing the daf; playing dhol → a musician playing the indian dhol; playing flute → a musician playing the flute; playing guitar → a musician playing the guitar; playing piano → a musician playing the piano; playing sitar → a musician playing the sitar; playing tabla → a musician playing the tabla; playing violin → a musician playing the violin; pole vault → an athlete jumps over the bar; pommel horse → gymnast performing pommel horse exercise; pull ups → a person doing pull ups on bar; punch → boxing match; push ups → push ups; rafting → group of people rafting on fast moving river; rock climbing indoor → rock climbing indoor; rope climbing → rope climbing; rowing → several people rowing a boat on the river; salsa spin → couple salsa dancing; shaving beard → young man shaving beard with razor; shotput → an athlete practicing shot put throw; skate boarding → a teenager skateboarding; skiing → skier skiing down; skijet → jet ski on the water; sky diving → sky diving; soccer juggling → soccer player juggling football; soccer penalty → soccer player doing penalty kick in a soccer match; still rings → gymnast performing on still rings; sumo wrestling → sumo wrestling; surfing → surfing; swing → kids swing at the park; table tennis shot → a person playing table tennis; tai chi → a person doing TaiChi; tennis swing → a person playing tennis; throw discus → an athlete practicing discus throw; trampoline jumping → trampoline jumping; typing → typing on computer keyboard; uneven bars → a gymnast performing on the uneven bars; volleyball spiking → people playing volleyball; walking with dog → walking with dog; wall pushups → a person standing, doing pushups on the wall; writing on board → a person writing on the blackboard; yo yo → a kid playing Yo-Yo.*

