# OpenReview forum: "Reuse and Diffuse: Iterative Denoising for Text-to-Video Generation"
_ICLR.cc/2024/Conference — ICLR 2024 Conference Withdrawn Submission_

### Official Review · Reviewer_DaXJ · 2023-10-24

**Soundness:** 2 fair
**Presentation:** 3 good
**Contribution:** 2 fair
**Rating:** 3
**Confidence:** 5

**Summary:**

This paper focuses on text-to-video generation. It proposes a pipeline to enrich existing video classification datasets with captions by leveraging LLMs to segment and caption videos. It also proposes an iterative text-to-video generation method to produce more video frames with limited computational resources. Experiments are conducted on UCF101 dataset.

**Strengths:**

1. The idea of using LLM to segment and caption videos from action recognition datasets can enrich existing video classification dataset with captions.
2. Experiment results on UCF101 demonstrate superior performance compared to previous methods.

**Weaknesses:**

1. The authors propose an iterative text-to-video generation method, which can generate arbitrary length videos. However, it is not evaluated in experiments. Besides, Many implementation details in their iterative generation pipeline are missing, such as the value of V_{max}, T, N, M, etc.

2. This paper focus on text-to-video generation, where video quality (IS, FVD) achieves the best performance on UCF101. However, the alignment between the input sentence and the generated video is not evaluated. Many previous papers report CLIPSIM on MSRVTT dataset to measure video-language alignment. It would be better to report this number and make a comparison. Besides, it would be better to have human evaluation metrics for different methods.

3. Since many classes between UCF101 and Kinetics700 are overlapped, it makes the comparison in Table 1 not fair enough. The authors can manually remove all classes which overlap with the Kinetics-400 dataset similar to the zero-shot action recognition settings[1].

4. the iterative generation idea is lack of novelty, similar idea has been proposed in [2].

[1] Doshi et al., Zero-Shot Action Recognition with Transformer-based Video Semantic Embedding, CVPR 2023 Workshop
[2] Harvey et al. Flexible Diffusion Modeling of Long Videos. NeurIPS 2022

**Questions:**

1.	What’s the strategy of prompt design in UCF101? How the new prompts improve the final performance?
2.	What’s the time complexity of this model? How long will it take to generate a video?
3.	What’s the difference between your iterative method and the method in [2]?

---

### Official Review · Reviewer_btaf · 2023-10-31

**Soundness:** 2 fair
**Presentation:** 4 excellent
**Contribution:** 2 fair
**Rating:** 5
**Confidence:** 5

**Summary:**

This paper proposed a new text-to-video generation method based on latent diffusion model (LDM) on text-to-image generation. The contribution of the proposed method has three folds: 1) A modified LDM UNet with temporal modeling modules for text-to-video generation. 2) A new training dataset for the text-to-video generation extended from a few image and video datasets. 3) A set of strategies for tackling auto-regressive long video generation including FNR, PNS, and DSG. The experimental results on UCF101 show that the proposed method outperforms the compared baselines.

**Strengths:**

- [writing] This paper is well-written. The content is easy to follow and all figures and tables are straight-forward.
- [method] The way of converting existing video and image datasets into text-to-video training datasets is insightful for other video generation works since high-quality diverse video datasets are expensive to obtain.
- [method] The auto-regressive long video generation strategy based on noise reshuffle and reuse is interesting.
- [experiment] The experimental results on UCF101 show that the proposed method can outperform the compared strong baselines in video quality metrics.

**Weaknesses:**

- [method] The text-to-video diffusion model based on LDM has been widely studied by Align-Your-Latent,  MagicVideo, and Latent-Shift, where the proposed temporal UNet shares a similar idea of those existing methods that use temporal convolution/attention, resulting in a lack of novelty in the architecture design.
- [method] For the noise reuse part, PNS and DSG are to directly share part of the noise of the already generated frames, which makes sense to me. However, FNR reverses the noise of the previously generated frames, which I think would contradict the temporal flow of the video. For example, for video frames f_n-2, f_n-1, f_n, f_n+1, f_n+2, letting f_n-2 and f_n-1 be already generated conditioned frames, in FNR, the conditioned frames for generating f_n, f_n+1, f_n+2 would be f_n-1, f_n-2 rather than f_n-2, f_n-1, where the temporal order is reversed. I think the authors should give more explanation here as to why the reversed order is better.
- [experiment] Only experimental results on UCF101 are provided, while other baselines compared the zero-shot MSRVTT performance, which I think this paper should also follow.
- [experiment] No user study comparison is presented.

**Questions:**

- In the comparison experiments on UCF101, most existing works only train the model on the webvid10M dataset, I wonder what is the training dataset used to produce the results in Table 1.
- I found the video generated by the proposed method is very flurry, especially the background. Do you have any explanation for it?

---

### Official Review · Reviewer_qJFK · 2023-11-01

**Soundness:** 2 fair
**Presentation:** 2 fair
**Contribution:** 2 fair
**Rating:** 5
**Confidence:** 4

**Summary:**

This paper proposes a framework called "Reuse and Diffuse" dubbed VidRD to generate consistent longer videos. By injecting some 3D temporal layers into LDM, an initial video clip can be generated by such temporal-aware LDM. The following frames are produced iteratively by reusing the latent features of the previous clip and imitating the previous diffusion process. For iterative generation, VidRD applies Frame-level Noise Reversion (FNR) to reuse the initial noise in reverse order from the previous video clip, a Past-dependent Noise
Sampling (PNS) scheme that brings a new random noise for the last several video frames, and a Denoising with Staged Guidance (DSG) strategy to refine temporal consistencies between video clips. For the autoencoder used for translating between pixel space and latent space, the authors inject temporal layers into its decoder and fine-tune these layers for higher temporal consistency. The authors also introduce a set of strategies for composing video-text data that involve diverse content from multiple existing datasets, including action recognition and image-text datasets.

**Strengths:**

- The introduced method is carefully elaborated to facilitate the readers' understanding. Sufficient details are provided.

- The relevant literature is well-discussed and organized.

- The proposed idea of reusing the latent features of the previous clip and imitating the previous diffusion process is generally interesting and brings certain insights.

**Weaknesses:**

- The reviewer's main concern is the actual experimental performance. By checking the provided video results, the temporal consistency can still be improved. Also, the generated motion still looks a bit simple in most cases. Some visual details can be further ameliorated. Providing more discussions and analyses on these could strengthen this paper further.

- Regarding the experiment evaluation, since the currently used metrics for evaluating video generation models are not fully reliable and may be inconsistent with human perception, a user study is highly recommended to be included to make the quantitative evaluation more reliable.

- The reviewer is a bit confused about the primary motivation of the paper. The paper just highlights the importance of producing more frames for long video generation at first, while some modules are also designed to enhance temporal continuity and quality. The authors also mentioned the efficient training of the proposed method but lacked the actual analysis.

- The reviewer is interested in the computational and memory cost of the proposed strategy as the authors also mentioned such cost is a formidable challenge for text-to-video generation. Providing more statistics, analyses, and discussions can be useful.

- Since the authors claim the video-text data composition as one of the contributions, the reviewer is interested in the accuracy of the captioning and the quality of the data. Providing more evaluation and analysis of the data and captions to demonstrate their merits is beneficial.

- The paper writing can be further improved. The current version is a bit hard to follow.

**Questions:**

- In the hyper-parameter ablation studies, what happens to the quantitative metrics and visual quality if the larger values of $\alpha$ and $\beta$ are used?

- It is advisable to include more qualitative examples in the paper or Appendix to showcase the effectiveness of the proposed VidRD over state-of-the-art baselines.

- Paragraph 2 of the Introduction section is too long, which can be separated into two paragraphs.

---

### Official Review · Reviewer_qhyu · 2023-11-01

**Soundness:** 2 fair
**Presentation:** 3 good
**Contribution:** 2 fair
**Rating:** 3
**Confidence:** 4

**Summary:**

This paper proposes a new architecture for text-to-video generation by diffusion models. Specifically, temporal guidance module is used to generate one video clip in a frame-by-frame way where the number of fixed random noise reduces as the temporal step grows. A temporal attention module is also interweaved into auto-encoder. By doing so, the generated video quality and smoothness improve a lot.

**Strengths:**

The way of introducing temporal guidance sounds reasonable and theorectically improve the smoothness of synthesized video. The illustration is easily-understood and the paper is well-written.

**Weaknesses:**

The visual quality of synthesized video is far away from current state-of-the-art text-to-video methods. Although FID is lower than other approaches, the smoothness and diversity of synthesized video is limited. More complex motion and scene generation is needed to support the superiority of this method.

**Questions:**

1. Why not try more complex prompt?
2. The smoothness of many scenes including train and owl is poor and I doubt the performance on more complex scenes.
3. Besides FID, is there any other evaluation metric helpful to demonstrate the superiority of video quality, expecially in video coherence and the fitness between text prompt and synthesized video?